# A Comprehensive Review and Update on Cannabis Hyperemesis Syndrome

**DOI:** 10.3390/ph17111549

**Published:** 2024-11-18

**Authors:** Priyadarshini Loganathan, Mahesh Gajendran, Hemant Goyal

**Affiliations:** 1Gastroenterology, UT Health San Antonio, San Antonio, TX 78229, USA; 2Gastroenterology, Borland Groover, Baptist Medical Center-Downtown, Jacksonville, FL 32207, USA

**Keywords:** cannabis, hyperemesis syndrome, hot shower, benzodiazepines, haloperidol, marijuana legalization, CBD, CBD receptors

## Abstract

Cannabis, derived from *Cannabis sativa* plants, is a prevalent illicit substance in the United States, containing over 400 chemicals, including 100 cannabinoids, each affecting the body’s organs differently upon ingestion. Cannabis hyperemesis syndrome (CHS) is a gut–brain axis disorder characterized by recurring nausea and vomiting intensified by excessive cannabis consumption. CHS often goes undiagnosed due to inconsistent criteria, subjective symptoms, and similarity to cyclical vomiting syndrome (CVS). Understanding the endocannabinoid system (ECS) and its dual response (pro-emetic at higher doses and anti-emetic at lower doses) is crucial in the pathophysiology of CHS. Recent research noted that type 1 cannabinoid receptors in the intestinal nerve plexus exhibit an inhibitory effect on gastrointestinal motility. At the same time, the thermoregulatory function of endocannabinoids might explain compulsive hot bathing in CHS patients. The prevalence of cannabis CHS is expected to rise as legal restrictions on its recreational use decrease in several states. Education and awareness are vital in diagnosing and treating CHS as its prevalence increases. This comprehensive review explores the ECS’s involvement, CHS management approaches, and knowledge gaps to enhance understanding of this syndrome.

## 1. Introduction

For thousands of years, cannabis and its derivatives, including hashish, have been utilized for their psychoactive properties [1]. It has been found that in 3500 BC, Romanian kurgans burned cannabis for ceremonial practices. Additionally, Israeli archaeologists discovered cannabis residues from the eighth-century B.C. shrine, indicating its use in ancient Jewish religious ceremonies [2]. Cannabis was first used for medicinal purposes around 400 AD in the United States (U.S.) [3]. Ancient Chinese texts document the use of *Cannabis sativa* for pain and cramp relief.

In 1850, cannabis was included in the U.S. Pharmacopeia, though due to federal restrictions, it was removed after the Marijuana Tax Act of 1937 [4]. In 1996, legalizing cannabis efforts began to gain traction, and California became the first state to legalize medicinal cannabis under the Compassionate Use Act. Currently (2024), per the Centers for Disease Control and Prevention (CDC), 47 states allow for the use of cannabis for medical purposes, and thirty-eight states allow for the use of cannabis for medical purposes through comprehensive programs. Nine states have medical programs that only allow CBD/low-THC products for qualifying medical condition(s) as defined by the state. (https://www.cdc.gov/cannabis/data-research/facts-stats/index.html, accessed on 4 September 2024). In the U.S., cannabis is the most misused drug, and over the past 30 years, our knowledge of cannabis has advanced significantly.

The effects of cannabis and its interactions on the various organ systems were elucidated further with the discovery of the endocannabinoid system (ECS) [1]. Tetrahydrocannabinol (THC) is the primary psychoactive compound in cannabis. The role of cannabinoid (C.B.) receptors has enhanced our understanding of cannabis’s effects on inflammation, seizures, nausea, and appetite regulation [5,6,7]. The ongoing cannabis legalization is expected to advance more research into its therapeutic potential.

Despite its medicinal benefits, cannabis use has increased adverse effects such as paradoxical hyperemesis, intoxication, and behavioral changes like anxiety and altered perceptions. Chronic cannabis use can result in dependency, affecting about 9% of users [8]. Cannabis has anti-emetic properties at low doses; however, at higher doses, it induces vomiting, known as cannabis hyperemesis syndrome (CHS) [9]. In Australia, the term CHS was first used in 2004, and since then, several cases of hyperemesis have been reported due to cannabis use [10]. The Rome IV diagnostic criteria for functional gastrointestinal disorders categorizes CHS as a disorder with episodic nausea and vomiting associated with heavy cannabis use.

Additionally, it is caused by disturbances in the gut–brain axis that do not have any other identifiable organic pathology [11]. Symptoms of CHS often improve with cessation of cannabis use but are also noted to respond with compulsive hot showers [12]. The exact prevalence of CHS is hard to assess due to its variable presentation. It is essential to raise CHS awareness with increasing cannabis legalization, which would also help us understand its mechanisms, early recognition, and treatment. This review will explore the recent update on CHS’s pathophysiology, prevalence, and treatment options based on the Rome criteria.

## 2. Pathophysiology

Historically, cannabis has been used to stimulate appetite and as an anti-emetic. The FDA approves its use for chemotherapy-induced nausea and vomiting when other anti-emetic treatments fail. Cannabis broadly affects the gastrointestinal system, affecting its secretions, appetite, inflammation, and motility [13,14,15]. Cannabis has over 100 cannabinoids in it and has varied effects and toxicity dependent on the THC-to-other-cannabinoids ratio [16]. Cannabinoids like anandamide (AEA) and 2-arachidonoylglycerol (2-AG) have shown anti-emetic effects in animals that can vomit (e.g., ferrets and shrews) and in those that cannot (e.g., rodents) [17,18]. In rodents, indirect measures such as taste aversion and facial expressions are identified to confirm the anti-emetic properties of cannabinoids [18].

Several studies have shown the anti-emetic properties of Cannabinoid and TRPV1 agonists (transient receptor potential cation channel subfamily V member 1, also known as the capsaicin receptor and the vanilloid receptor 1) [19]. However, chronic cannabis use may lead to CHS demonstrating its complex dual effects [10,20].

Cannabis’s varied potency accounts for its biphasic effects, such as anti-emetic properties at low doses and pro-emetic at higher doses [21]. G.I. disorders affecting the gut–brain axis, such as CHS, irritable bowel syndrome, and functional dyspepsia, are generally due to psychological disturbance associated with heightened visceral sensitivity, autonomic dysfunction, and altered gastric emptying [20]. Several mechanisms are delineated as the cause of CHS, though in many cases, it is due to multiple contributing factors, as described below.

### 2.1. Pituitary–Adrenal Axis

Cannabinoids affect the pituitary–adrenal axis and stress-responsive brain regions. Studies suggest that CHS may involve disruption at the hippocampal–hypothalamic–pituitary level [22]. Chronic cannabis use can lower pituitary hormone levels, including the growth hormone, follicle-stimulating hormone, and luteinizing hormone, which has been shown to normalize after stopping use [23,24].

### 2.2. Endocannabinoid System (ECS)

Understanding the ECS and its impact on the brain’s vomiting center is essential in the CHS pathophysiology [25,26]. The ECS includes ligands, receptors, signaling pathways, and enzymes acting as regulators and inhibitors. This includes cannabinoids like AEA and 2-AG, their synthesizing and degrading enzymes, and receptors CB1 and CB2, which are crucial for understanding Cannabis’s biphasic effects [27].

i.Ligands:

The ligands in ECS are generated in response to stress and bind to their specific receptors. There are two types of ECS ligands: endogenous and exogenous.

The endogenous ligands, AEA and 2-AG, are derived from arachidonic acid. 2-AG is mainly located in the brain and is primarily involved in the signaling process. AEA and 2-AG are produced from cell membrane lipids and move to the extracellular space via diffusion, endocytosis, carrier transport, translocation, and possibly heat shock proteins [28]. They stimulate the cannabinoid receptors in various brain regions, including the temporal lobe, orbitofrontal cortex, insula, and parahippocampal areas, to produce their effects [29].

Exogenous ligands, such as N-acyl ethanolamines and mono-acyl-glycerols, include notable compounds like THC (which contains a dibenzopyran ring), cannabidiol, cannabigerol, and cannabinol [23,30]. These ligands interact with G protein-coupled receptors (GPR), GPR18 and GPR55, peroxisome proliferator-activated receptors (PPARs), and TRPV1.

ii.Receptors:

CB1 and CB2 are the two primary receptors in ECS. TRPV1, PPARα, GRP55, and GRP119 are the other receptors influenced by cannabinoids [27]. Cannabinoid receptor type 1 (CB1R) is a G protein-coupled receptor (GPCR) primarily expressed in the central nervous system but also found in peripheral tissues. CB1 receptors affect gastric secretion, motility, inflammation, and sensation. They suppress the hypothalamic–pituitary–adrenal axis and sympathetic systems when activated. They are in the cerebral cortex, anterior cingulate gyrus, hippocampus, cerebellum, and basal ganglia. In the gastrointestinal system, they are present on both intrinsic and extrinsic neurons of the enteric nervous system [31]. During the last decade, the discovery of CB1R allosteric modulators has provided new tools to target the CB1R [32]. The CB1 receptor is a valuable target for treating a wide range of disorders, including anxiety, pain, and neurodegeneration. However, the development of drug candidates for CB1 is challenged by side effects, rapid tolerance buildup, and the risk of abuse [33].

CB2 receptors help to control inflammation, visceral pain, and intestinal motility [34]. They are found in lamina propria plasma cells and activated macrophages. Nausea and vomiting are regulated in ECS through central and peripheral pathways [35]. Dysfunction in these pathways leads to recurrent nausea and vomiting in CHS. Potential causes of CHS include influence on the activity of cannabinoid receptors, conversion of cannabis into emetic substances, or contamination with other toxins [36]. Cannabis leads to upregulation of CB1 receptor activity in the hypothalamus, which enhances the hypothermic effects of THC.

The effects of cannabis are biphasic: low doses tend to reduce nausea, while high doses can induce vomiting [37]. This is explained by the partial agonist action of Delta-9-THC at CB1 receptors at low doses. In contrast, at high tissue concentrations, chronic use produces an antagonistic effect, potentially resulting in withdrawal symptoms like vomiting [38]. This antagonistic effect increases the release and turnover of emetogenic transmitters such as serotonin, dopamine, and substance P [39,40]. At high doses or with chronic cannabis use, CB1 receptor activation can paradoxically trigger the symptoms of CHS. This dual effect is partly attributed to receptor desensitization, internalization, and dysregulated signaling [41,42]. These processes could reduce the effectiveness of endocannabinoid feedback inhibition, leading to increased excitatory activity in the brainstem or gastrointestinal system, thereby resulting in hyperemesis [43].

iii.Enzymes:

AEA and 2-AG are produced from membrane lipids and are crucial in ECS [27]. Diacylglycerol (DAG) is broken down by DAG lipase to generate 2-AG, while N-acyl phosphatidylethanolamine is broken down by phospholipase-D to produce AEA [31]. Both processes involve a calcium-sensitive rate-limiting step. AEA and 2-AG are inactivated intracellularly: 2-AG is mainly hydrolyzed by monoacylglycerol lipase (MAG), found in the mucosa and muscle layers of the duodenum, ileum, and both the proximal and distal colon [44]. AEA is primarily degraded by the fatty acid amide hydrolase (FAAH) pathway, located in the myenteric plexus throughout the gut [45,46]. Since AEA levels decrease during stress, increasing its levels by inhibiting its degradation could offer a potential therapeutic approach to nausea and vomiting. Figure 1 explains this process.

iv.Genetic predisposition:

Both CVS and CHS are complex gastrointestinal conditions influenced by several entities, including genetic, environmental, and lifestyle factors. While their genetic underpinnings are still not fully understood, research has suggested potential genetic predispositions for each. The CVS has strong links to mitochondrial dysfunction and neurobiological pathways related to migraine, while CHS is primarily influenced by chronic cannabis use and endocannabinoid system dysfunction. Understanding these primary differences in the pathophysiology between these two disease entities is crucial for clinicians when diagnosing, especially since they share overlapping gastrointestinal symptoms.

A small dataset study showed five mutations with plausible etiological roles in the phenomenology of CHS symptoms and signs. These genes are COMT, transient receptor potential vanilloid receptor 1 (TRPV1), CYP2C9, the gene coding for the dopamine-2 receptor (DRD2), and the ATP-binding cassette transporter gene (ABCA1). This constellation of genetic susceptibilities may represent a valid diagnostic tool for identifying at-risk individuals. It is important to note that CHS is not a “functional” G.I. disorder but rather a manifestation of the gene–environment interaction in a rare genetic disease unmasked by a toxic reaction to excessive THC exposure [47]. A recent study Omri Bar et al. showed 12 genes that were “Highly likely” (SCN4A, CACNA1A, CACNA1S, RYR2, TRAP1, MEFV) or “Likely” (SCN9A, TNFRSF1A, POLG, SCN10A, POGZ, TRPA1) to be CVS-related [48]. As per this study, CVS is likely the result of a vicious cycle of elevated intracellular cations and mitochondrial dysfunction leading to cellular hyperexcitability [48].

### 2.3. Sympathetic Dysregulation

Sympathetic and parasympathetic systems play interlinked roles in emesis. The chemoreceptor trigger zone sends signals via the efferent vagus nerve, triggering responses in the parasympathetic and sympathetic nervous systems. This activation leads to the emetic reflex, which includes increased salivation, deep breathing, glottis closure, pyloric sphincter relaxation, retroperistalsis, and abdominal muscle contraction. CHS may involve dysfunction in the sympathetic nervous system [49]. This is evidenced by symptoms like rapid heartbeat, sweating, hot flashes, high blood pressure, and tremors, often during the hyperemesis phase [49].

### 2.4. Stress

Psychological stress, such as post-traumatic stress disorder or a history of physical and sexual abuse, are potential triggers for disrupting the expected anti-emetic effects of THC. Though the precise mechanisms remain unclear, higher amounts of marijuana consumption, genetic influences, and psychological stress lead to intoxication and paradoxically promote vomiting.

Cannabinoids have a strong affinity for fat and accumulate in cerebral fat, acting as a reservoir of THC in adipose tissue. The sympathetic nervous system becomes activated during starvation or heightened physical stress. This results in elevated adrenocorticotropic hormone (ACTH) action on adipocyte receptors, causing lipolysis to meet bodily demands [50]. Lipolysis from elevated ACTH triggers the release of THC from fat cells. This leads to high THC concentrations, particularly after consuming potent cannabis, which explains its pro-emetic effects [51]. Furthermore, chronic marijuana use has been shown to impair gastric emptying, thereby causing nausea and vomiting after meals [52]. This was demonstrated in a study by McCallum et al., where male participants were given either marijuana or a placebo before undergoing a radionuclide gastric emptying test [53].

## 3. Clinical Presentation

The Rome IV criteria categorize functional nausea and vomiting disorders into three types: chronic nausea and vomiting syndrome, cyclic vomiting syndrome (CVS), and CHS. The main symptoms of CHS include repetitive vomiting episodes occurring in individuals with chronic, daily cannabis use, with relief of symptoms following the cessation of cannabis use. It is crucial to differentiate CHS from CVS for appropriate management (Table 1). Many times, CHS presents as epigastric abdominal pain, often accompanying nausea and vomiting, though Rome IV does not include it. The vomiting and abdominal pain are suppressed by hot showers, possibly due to their relaxation and distraction effects. There are periods of well-being or remission lasting from days to weeks between the symptoms episodes. However, attacks may become more frequent over time if there is continued usage of cannabinoids.

CHS involves 3 phases: prodromal, hyperemesis, and recovery.

### 3.1. Prodromal Phase

The prodromal phase can be present for several months. During this phase, patients may experience morning nausea, abdominal discomfort, or anxiety about vomiting. Despite these G.I. symptoms, patients often eat well, maintain weight, and remain functional at work. The patients continue using cannabis in this phase, believing in its anti-nausea effects.

### 3.2. Hyperemesis Phase

The hyperemesis phase can last for several days. This phase begins with severe symptoms that intensify rapidly within a few hours [54]. Patients present with distressed stomach, intense, persistent nausea, and frequent vomiting, feeling as though a relapse is imminent in this phase. This episode is debilitating and overwhelming, with patients vomiting and retching up to five times per hour, requiring several emergency room (E.R.) visits. Abdominal pain generally starts in the epigastric region and progresses to more diffuse abdominal pain. The intense diffuse abdominal pain may sometimes need extensive diagnostic workup, including biliary scans to rule out acute cholelithiasis or choledocholithiasis and multiple computed tomography (C.T.) imaging to rule out acute abdominal conditions, including pancreatitis. A sympathetic overactivity during this phase results in symptoms such as tachycardia, hypertension, hot flashes, sweating, and trembling [42]. Due to excessive nausea and vomiting, patients are often found to have hypokalemia, volume depletion, acute renal failure, hypophosphatemia, and mild reactive leukocytosis [55,56,57]. Multiple and forceful vomiting events can cause Mallory–Weiss tears with hematemesis and rarely lead to pneumomediastinum or Boerhaave’s syndrome [58].

### 3.3. Recovery Phase

In this phase, patients gradually resume normal eating and dietary habits. Patients experience complete relief of the symptoms, which can last days, weeks, or even months. The duration of this phase ranges from weeks to months, depending on resuming marijuana use, which may trigger another relapse. Throughout this phase, the patient maintains an average weight and returns to their baseline state [49].

### 3.4. Pathological Bathing Behavior

Several previous studies have described the characteristics of frequent and prolonged hot shower use common among patients with CHS. Patients often adopt this behavior to alleviate nausea, vomiting, and abdominal pain symptoms of CHS, and some reports have referred to this symptom as CHS as “cannabis hot shower syndrome”. It is hypothesized that hot showers help stabilize the thalamic thermostat, which is frequently disrupted by chronic cannabis use, including CHS. Often, they are used as a self-treatment in CHS. However, this proposed mechanism has not been empirically validated [59]. Though many patients with CHS may use hot bathing or showering to obtain relief from its symptoms, more than 10% may not exhibit this behavior [60].

Additionally, similar patterns of hot shower behaviors are observed in cyclic vomiting syndrome (CVS), as well as in preadolescents and adolescents with no history of cannabis use [61]. Thus, hot showers may be associated with CHS; they are not a unique diagnostic feature of CHS and are not included in the Rome IV diagnostic criteria [62]. CHS has more male predominance and, similar to CVS, primarily affects young people.

## 4. Diagnosis

Failure to recognize this disorder can result in multiple E.R. visits and extensive recurring serum testing and imaging evaluations with increased healthcare-related costs. It is crucial to exclude other entities such as Addison’s disease, migraines, hyperemesis gravidarum, bulimia, and psychogenic vomiting, which can mimic CHS symptoms and may also occur alongside it. A thorough medical history, complete physical examination, and focused diagnostic testing help differentiate from these other differential conditions. CHS is classified as a type of functional gut–brain disorder and a variant of cyclic vomiting syndrome (CVS) per the Rome IV structured framework. However, it is essential to differentiate CHS from CVS. Venkatesan et al. [21] described the features of CHS, including clinical features, cannabis use patterns, and symptom resolution after at least six months of abstinence, helping differentiate it from CVS. However, uncertainties remain about cannabis dosage, individual and genetic factors, the duration of abstinence, and the role of abdominal pain in its diagnosis.

In patients with CHS, elevated urinary concentrations of the cannabis metabolite carboxy-THC (THC-COOH) exceeding 100 ng/mL are indicative of significant chronic cannabis exposure.

## 5. Management

The management of CHS largely relies on the severity of symptoms, the emergence of complications, and measures to prevent future recurrence. Evidence-based management of CHS is based on case series and small clinical trials [63]. The recent 2024 American Gastroenterology Association (AGA) clinical practice update recommended combining evidence-based psychosocial interventions and pharmacological treatments for the successful long-term management of CHS [63]. Table 2 outlines the treatment options for CHS.

### 5.1. Acute Treatment Strategies

CHS patients present to the emergency department (E.D.) during the hyperemesis phase. Complications of CHS may include acute renal failure, hypokalemia, hypophosphatemia, esophageal injuries such as Mallory–Weiss tear, and pneumomediastinum. The primary treatment objectives are intravenous hydration and correction of electrolyte imbalances. Repeated abdominal imaging and extensive laboratory tests, in most instances, yield inconclusive results. Conventional anti-emetics, such as ondansetron and promethazine, are routinely utilized in the acute symptomatic phase [42]. A systematic review by Richards et al. [64] showed that these standard anti-emetics are often ineffective when used alone and demonstrated superior efficacy with intravenous benzodiazepines.

### 5.2. Abortive Therapies

i.Benzodiazepines:

Benzodiazepines, such as lorazepam, have proven acute treatment for CHS [65,66]. Intravenous lorazepam administered in doses of 1 to 2 mg every 4 to 6 h has shown symptom relief [65,66]. Patients may also benefit from oral lorazepam tablets, doses between 0.5 to 1 mg every 6 to 8 h on discharge. Benzodiazepines, with their gamma-aminobutyric acid (GABA) agonistic actions, inhibit the medullary and vestibular nuclei, causing anti-emetic action. Additionally, anxiolytic and sedative properties aid in counteracting the abnormal sympathetic nervous system response, helping in the reduction in vomiting and decreasing pain perception [67].

GABA agonists influence the G.I. tract by decreasing G.I. motility, mucosal hemostasis, and the release of chemical mediators such as histamine, prostaglandin, acetylcholine, and serotonin [68]. Cannabinoid’s interaction with cannabinoid receptors inhibits GABA-mediated neurotransmission, thus reducing the negative inhibition of dopaminergic neurons. This increases dopamine release and decreases extracellular glutamate in the striatum and mesolimbic systems [69,70]. These physiological alterations manifest as anxiety, tremors, and paranoia in some cannabis users.

Abrupt cessation of cannabis use may cause catatonia from hypoactivity of GABA and dopamine D2 receptors, along with hyperactivity of the glutamate N-methyl-D-aspartate receptor [71,72]. These situations can be effectively treated with benzodiazepines. Nevertheless, it is essential to be aware of the adverse effects of benzodiazepine, such as oversedation, hypoventilation, dizziness, confusion, incoordination, and the long-term effects of addiction.

ii.Droperidol:

It has anti-emetic and antipsychotic properties from the dopamine antagonist effect. A systematic study by Furyk et al. showed that intravenous droperidol of doses ranging from 0.625 to 2.5 mg was the only treatment that had statistically significant improvements (*p* < 0.05) on the visual analog scale compared to a placebo in 48 CHS patients [73].

Comparative studies have demonstrated shorter hospital stays, decreased reliance on other anti-emetics, and significantly reduced nausea severity from the baseline in dronabinol-treated CHS patients relative to the placebo [74]. The study by Lee et al. showed that the median hospital stay for the droperidol-treated group (37 patients) was significantly lower than that of the untreated group (39 patients) (6.7 h vs. 13.9 h, *p* = 0.014) [74]. Nonetheless, this retrospective study by Lee et al. was associated with significant biases in participant selection and result reporting [74].

iii.Haloperidol:

It is an off-label anti-emetic agent [75]. It is mainly utilized for treating agitated patients and causes sedative effects at doses of 2 to 10 mg intravenously, with a maximum daily dosage of 30 mg [76]. Haloperidol is a D2 receptor antagonist that acts within the mesolimbic and mesocortical pathways. Its sedative and anti-emetic properties help to manage hyperemesis in CHS patients. It is generally administered at 0.5 to 2 mg intravenously every six hours, as needed [77,78]. Additionally, interactions between dopamine and CB1 signaling pathways may contribute to haloperidol’s effectiveness in treating CHS [79].

Witsil and Mycyk reported a case series where 5 mg of intravenous haloperidol alleviated nausea and vomiting in patients presenting to the emergency department [80]. A recent randomized controlled trial by Ruberto et al. [81] found that haloperidol at 0.05 or 0.1 mg/kg was more effective than ondansetron in reducing nausea and vomiting according to the visual analog scale (*p* = 0.01). Additionally, a shorter stay in the emergency department was seen in the haloperidol group (*p* = 0.03) compared to ondansetron. However, dystonia was observed in two patients with higher doses of haloperidol [81]. A recent meta-analysis assessed 492 patients on the efficacy of capsaicin cream (5 studies, n = 386) and dopamine antagonists (2 studies, haloperidol, droperidol; n = 106) in CHS [82]. There was mixed evidence for the efficacy of capsaicin over dopamine antagonists in reducing nausea and emesis. However, dopamine antagonists were more beneficial than standard care or no treatment [82].

It is imperative to recognize the adverse reactions of haloperidol, such as extrapyramidal reactions, Parkinsonism, dystonia, and QTc prolongation [83]. The intravenous administration of haloperidol is associated with dose-dependent QT prolongation at doses exceeding 2 mg [84]. A study demonstrated that 513 patients per 100,000 teenage cannabis users were found to have long QT syndrome [75]. Caution must be implemented during Haloperidol treatment as underlying QT prolongation may worsen it in this patient population.

iv.Aprepitant:

Aprepitant is a Neurokinin 1 Receptor (NK1) antagonist [85]. It blocks NK1 and inhibits the binding of substance P, thereby preventing receptor activation and reducing nausea sensation in the brainstem [85]. Aprepitant can be considered as third-line management when Lorazepam or Haloperidol fails, owing to its efficacy in treating moderate to severe CVS.

v.Scopolamine patch:

Scopolamine patches are an antimuscarinic agent with an anti-emetic effect for up to three days, ensuring consistent absorption between oral and parenteral medications. Scopolamine is commonly used to alleviate nausea and motion sickness. CHS patients with severe vomiting who are unable to retain their oral anti-emetic medications can benefit from a scopolamine patch.

### 5.3. Abdominal Pain Management

Parenteral narcotics are contraindicated for abdominal pain from CHS, as these drugs may exacerbate hyperemesis and lead to addictive behavior. Abdominal pain management in CHS should focus on treatment that avoids G.I. side effects or addiction potential.

i.Capsaicin:

Capsaicin, a topical agent with an active compound derived from chili peppers, interacts with transient receptor potential vanilloid-1 (TRPV1) receptors [86]. TRPV1 receptors are involved in the modulation of transmitting pain signals and altering pain perception [87]. These TRPV1 receptors are present throughout the gastrointestinal (G.I.) tract and the medullary vomiting center. They are frequently located closer to CB1 receptors, indicating a potential functional interaction. Upon topical application as a cream to the abdomen, capsaicin causes a sensation of heat at the application site, suppressing the underlying abdominal pain. A novel pilot randomized controlled trial by Dean et al. [88] showed that topical 0.1% capsaicin reduced nausea from the baseline by 46% at 60 min, compared to 24.9% in the placebo topical cream group. Additionally, capsaicin’s anti-emetic effect was more effective at 60 min than 30 min after the first application [88]. Significant improvements in nausea and vomiting, as well as shorter length of hospital stay, were noted in patients treated with 0.075% topical capsaicin applied to the abdominal region [62].

A retrospective cohort study of 43 emergency department (E.D.) patients showed that topical abdominal capsaicin application ranging from 0.025% to 0.1% reduced the median length of stay (LOS) by 22 min [89]. Patients who received capsaicin application needed fewer additional pain medications, such as opioids (*p* = 0.015) [89]. Administering 0.025% to 0.15% topical capsaicin cream earlier upon arrival reduced LOS in the E.D. (4.83 h compared to 7.09 h, *p* = 0.01) [90]. The FDA had recently approved an 8% capsaicin patch for managing diabetic peripheral neuropathic pain in the foot. This treatment may also help manage abdominal pain associated with CVS or CHS. Skin irritation and blistering at the application site were the most commonly encountered adverse effects of this topical application.

iiHot shower:

Pathological hot bathing can temporarily relieve CHS symptoms [64]. TPRV1 receptors are activated at temperatures above 43 °C attained during hot showers.

iii.Lidocaine:

Carnett’s sign refers to pain elicited by tensing the abdominal muscles or performing a straight leg raise. Lidocaine patches have been proposed as a means to relax the rectus muscle, potentially alleviating abdominal pain during acute flares [91].

### 5.4. Long-Term Strategies

i.Cessation of Cannabis:

Discontinuation of cannabis use in any form is required for complete long-term management of CHS. A multimodal approach, including structured psychotherapy such as cognitive behavior therapy (CBT), along with addiction counseling in educating patients about the consequences of cannabis use, is necessary [92]. Some patients may require rehabilitation programs to monitor the patient’s progress, ensure treatment adherence, and offer therapeutic support to achieve and maintain recovery. Mutual-help groups such as Marijuana Anonymous are beneficial to patients without access to structured programs.

The primary modes of psychosocial intervention in cannabis use disorder are CBT and motivational approaches, which include the importance of the individual or the social environment. More specifically, CBT and relapse prevention approaches primarily focus on the identification and management of thoughts, as well as external triggers, that lead to its use. These approaches teach coping and problem-solving skills and promote the substitution of cannabis-related behaviors with healthier alternative behaviors [93]. In contrast, motivational interviewing attempts to build motivation in an empathic and non-judgemental environment and emphasize the importance of self-efficacy and positive change. This approach is often enhanced by personalized feedback and education regarding the treatment seeker’s patterns of cannabis use, becoming motivational enhancement therapy. CBT and motivational approaches can be provided individually or in groups [93]. Besides these approaches, secondary options such as mindfulness-based meditation and drug counseling are highly beneficial. Mindfulness-based meditation is a new approach that promotes inner reflection and acceptance of experiences and negative effects by enhancing present-moment awareness and thus decreases the impact of triggers of use [93]. The holistic management options would provide a more comprehensive approach to long-term care.

In patients who do not improve with psychosocial interventions, gabapentin may be used as an adjunctive treatment to aid in addiction recovery. Cannabis withdrawal commonly presents with symptoms such as loss of appetite, anxiety, depression, physical tension, and insomnia, which increases the difficulty in ceasing cannabis use [93]. Amitriptyline and lorazepam can be used to mitigate withdrawal effects of cannabis. Furthermore, this process requires considerable effort and motivation from the patient. The duration to achieve complete recovery from cannabis use disorder may range from a minimum of 3 months to up to 4 years [94]. This variability in recovery time is partly attributable to its accumulation in adipose tissue and due to the extended half-life of THC.

ii.Tricyclic antidepressants (TCA):

TCAs are anticholinergics that modulate alpha-2-adrenoreceptors, thereby decreasing sympathetic nervous system activity and mitigating brain–gut autonomic dysfunction [95]. Amitryptine helps to prevent vomiting cycles in CHS, usually at doses between 50 and 200 mg daily [96]. TCA can be initiated during the acute episode in the E.D. or hospital, and dosage titration can be made during closer outpatient care. A study evaluating the TCA effect on CVS with CHS indicated that both conditions showed significant pain relief [5]. The CHS group achieved a 70% improvement, and the CVS group achieved an 80% improvement following the treatment with amitriptyline [5].

Amitriptyline is initially started at a low dose of 10 mg at night and gradually increased to 10 mg every 1–2 weeks until the therapeutic effect is achieved [96]. Slow up-titration helps to adapt and minimize anticholinergic side effects, including dry mouth, sedation, constipation, postural hypotension, palpitations, chronic fatigue, blurred vision, nightmares, and mild hallucinations. Additionally, gradual titration of the dose prevents cardiac arrhythmias. TCA is used in caution with underlying cardiac arrhythmias, recent myocardial infarction, mania, or severe liver disease [97]. Amitriptyline use is not advised during pregnancy, and it is classified as a Category C drug by the FDA.

Alternative TCAs such as nortriptyline and doxepin may be used when amitriptyline is not tolerated well, owing to its excessive sedative nature in certain patients. Nortriptyline and doxepin have fewer adverse effects and provide substantial therapeutic benefits. Doxepin is generally better tolerated and is started at a dose of 10 mg. The dose can be gradually increased in 10 mg increments every 1–2 weeks until the G.I. symptoms resolve without additional side effects. Cardiac arrhythmias have not been observed with gradual dose titration.

The combined approach of cannabis use reduction within 3–6 months, along with TCA, helps in preventing CHS episodes. The amitriptyline effect on CHS is significantly lowered in patients with continued usage of cannabis products. Once the patient has maintained CHS remission, defined as the absence of CHS attacks for 6–12 months while on TCA, its dosage can be gradually decreased by 10 mg per month.

iii.Management of mood disorders:

Mood disorders such as anxiety and depression often coexist in patients with CHS [98]. Many times, presentation during the hyperemesis phase may be similar to panic disorder. This similarity could be the reason why CHS tends to respond well to benzodiazepines, especially during these acute episodes. It is essential to treat the underlying mood disorder to achieve cannabis discontinuation and CHS remission. The Hamilton Rating Scale for Anxiety is commonly used to assess anxiety, while the Zung Depression Inventory is employed to evaluate depression severity [99,100].

Most patients with CHS attribute their vomiting cycles to associated intense stress and anxiety. A combination of pharmacotherapy and psychotherapy is generally more effective than either approach alone in treating the underlying mood disorder [101]. Selective serotonin reuptake inhibitors (SSRIs), serotonin-norepinephrine reuptake inhibitors (SNRIs), atypical antidepressants, and TCAs are the commonly used antidepressant aids in achieving remission in CHS. Duloxetine (an SNRI) and mirtazapine (an atypical antidepressant) are also frequently prescribed. Buspirone is effective for generalized anxiety disorder, but it may take 4–6 weeks to reach its full effectiveness [102]. Lorazepam is typically used in acute abortive therapy in CHS and may, at times, be needed as adjunctive therapy to treat underlying mood disorders [103].

iv.Nutrition:

In hospitalized patients with CHS during the hyperemesis phase, a “nothing by mouth” regimen and IV hydration are typically employed until symptoms improve. As recovery progresses, patients are initially given clear liquids and gradually advance to a regular diet as tolerated. In patients treated at home, recommendations are emphasized to consume fluids containing glucose and electrolytes between vomiting episodes to ensure adequate hydration. CHS patients generally do not experience significant weight loss, as periods of regular oral intake often compensate for the days of vomiting.

### 5.5. Recent Advancements

In the last decade, several reports describing the structure and function of the CB1R, its allosteric ligands, and their translational potential have increased enormously. With these new advancements, the application of site-directed mutagenesis, together with advanced physical methods (NMR, EPR, MS, FRET, and X-ray crystallography) and computational modeling, have improved the understanding of the complexity of the structure, function, and activity of cannabinoid receptors [32].

The orthosteric ligands of the CB1R were considered to be potential pharmaceuticals in the treatment of disorders such as drug addiction, obesity, and pain [33]. However, cannabinoid receptor activation results in adverse psychoactive effects (including depression and suicidal thoughts), which is concerning for them in clinical use [104]. The recent discovery of CB1R allosteric modulators described so far can be classified as indole derivatives (e.g., Org27569, Org29647, and Org27759), urea derivatives (e.g., PSNCBAM-1), endogenous ligands (lipoxin A4 and pregnenolone), and other compounds like synthetic cannabinoids (e.g., JHW007; a synthetic cannabidiol (CBD) or RTI-371, a tropane derivative) [104]. With more research, the complexity of allostery can be elucidated, which will be beneficial in the development of safe and efficacious drugs with no neuropsychiatric side effects.

## 6. Long-Term Outcome

Two treatment patterns were noted in Sifuentes et al.’s long-term follow-up study of CHS patients [105]. Some patients require a gradual increase in their maintenance dose to maintain stability, as dose tolerance leads to ‘breakthrough’ vomiting episodes. Once patients achieve stability with TCA therapy, evidenced by no emergency department visits for at least one year, the amitriptyline dosage can often be tapered or discontinued entirely over the following year. Female patients are frequently motivated to taper off amitriptyline in anticipation of pregnancy. This study’s findings indicate that over 40% of patients discontinue all treatments [105].

Long-term management success relies on patient and physician commitment, availability, and the coordination of regular follow-up visits. Establishing trust and rapport between the patient and physician is crucial. A good patient–physician relationship helps achieve CHS remission sooner and avoids unnecessary diagnostic workups [106]. Studies indicate that when patients trust their physicians, they are more likely to disclose sensitive health-related behaviors and adhere to medical recommendations [107]. This trust also encourages patients to accept a CHS diagnosis, preventing them from seeking unnecessary medical consultations and receiving inappropriate treatments.

## 7. Role of Public Health

Due to the rise in CHS prevalence, likely from cannabis legalization in more states, public health outreach programs can play a crucial role in bringing awareness and prevention to this condition. More educational campaigns targeting heavy recreational and medicinal cannabis users can help in recognizing and preventing CHS. Education focusing on early warning signs could prevent the condition from worsening. Partnership with cannabis producers and retailers to include CHS warnings on packaging similar to alcohol and tobacco products will improve awareness among its users. With more online usage among youngsters, social media campaigns and videos can spread information and awareness of CHS in this population. Public health can aid in creating safe consumption guidelines on lower dose usage for patients using it for medicinal reasons. Policy emphasizing mandatory labeling of high-potency cannabis products with information on their risk, including CHS, will benefit chronic users. Cannabis legalization should include balanced details on its benefits and potential risks. As cannabis becomes legal in more places, it is essential to monitor CHS cases across the country, which can help with public health strategies and policy decisions.

## 8. Conclusions

The prevalence of cannabis use disorder is expected to rise as legal restrictions on its recreational use decrease in several states. Nearly 20 years after CHS’s first report, current acute and long-term treatment strategies remain unfamiliar to many practitioners. Education and awareness, especially among E.R. and primary care providers, are vital in diagnosing and treating CHS as its prevalence rises. This will provide more data on CHS and facilitate the development of targeted novel therapeutic interventions for this condition in the future. Also, future longitudinal research exploring genetic predisposition and biomarkers could aid in diagnosing and treating CHS. It is also essential to study the public health implications of the legalization of cannabis in different states and its impact on healthcare utilization.

## Figures and Tables

**Figure 1 pharmaceuticals-17-01549-f001:**
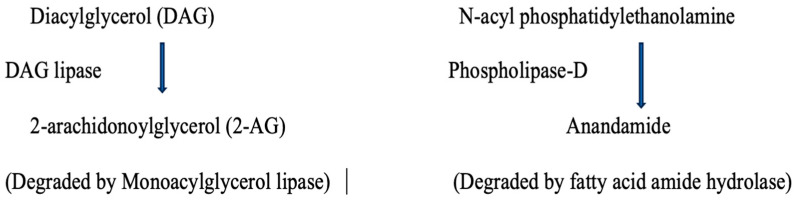
Enzymes in the endocannabinoid system.

**Table 1 pharmaceuticals-17-01549-t001:** Criteria for cyclic vomiting syndrome and cannabinoid hyperemesis syndrome.

Cyclic Vomiting Syndrome (CVS)	Cannabinoid Hyperemesis Syndrome (CHS)
Stereotypical episodes of vomiting acute in onset At least three discrete episodes in the prior year and two episodes in the past six months, occurring at least one week apart Absence of vomiting between episodes, but other milder symptoms can be present between cycles Criteria fulfilled for the last three months with symptom onset at least six months prior to diagnosis	Symptoms present for past three months with onset at least 6 months prior Stereotypical episodes lasting <one week At least 3 episodes in last one year and 2 episodes in last six months (occurring at least one week apart) No vomiting between episodes Milder symptoms can be present during this All these criteria should be associated with chronic use of cannabis and stop after its cessation

**Table 2 pharmaceuticals-17-01549-t002:** Management of cannabis hyperemesis syndrome.

Therapy	Mechanism and Advantages	Adverse Effects
Benzodiazepines	Useful for their anti-anxiety and anti-emetic effects and inhibition of the vestibular system	Sedation, altered consciousness
TCA	Reduce cholinergic neurotransmission and modulate alpha-2-adrenoreceptors, thereby decreasing sympathetic nervous system activity and mitigating brain–gut autonomic dysfunction	Arrhythmias
Anti-dopaminergic: Haloperidol Droperidol	Haloperidol is a broad-spectrum antiemetic. May interfere with CB1 signaling. Blockage of dopamine at the chemoreceptor trigger zone.	Arrhythmias, central nervous system side effects Dysrhythmias (Q.T. prolongation), oversedation
Dopaminergic agents: Promethazine Prochlorperazine	Effect CTZ area in the brain stem. Variable success noted	Arrhythmias, extrapyramidal effects, hypotension, and sedation related effects
Serotonergic antagonists: Ondansetron	First-line agents used for emesis. Variable response noted	Arrhythmias
Corticosteroids	Rarely used with limited response	Hyperglycemia and psychosis
Capsaicin	Bind to TRPV1 receptors in proximity to CB1	Skin irritation
Aprepitant:	Blocking NK1 receptors in the dorsal vagal complex of the brainstem inhibits the binding of substance P, thereby preventing receptor activation and reducing nausea sensation	
Volume repletion	Prevents dehydration related symptom	Minimal
Cannabis cessation	Required for long-term management	Patient compliance

TRPV1, transient receptor potential cation channel subfamily V member 1; TCA, tricyclic antidepressant; CB, cannabinoid; CTZ, chemoreceptor trigger zone.

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
