# Peer review of "A Comprehensive Review and Update on Cannabis Hyperemesis Syndrome"

_pharmaceuticals, 2024, doi:10.3390/ph17111549_

Round 1

Reviewer 1 Report

Comments and Suggestions for Authors

he manuscript provides a thorough review of Cannabis  and Cannabis Hyperemesis Syndrome (CHS), from its historical background, pathophysiology, clinical presentation, and management strategies. It offers a comprehensive summary, supported by various studies, which makes it a valuable resource for clinicians and researchers interested in CHS and its associated challenges. However, there are areas that require further  improvement to enhance quality of the manuscript.

1. the endocannabinoid system (ECS) could benefit from further clarification, especially regarding how CHS contrasts with the antiemetic effects of cannabis at lower doses. The manuscript mentions biphasic effects (antiemetic at low doses, pro-emetic at higher doses), but this concept could be expanded with more details about the molecular mechanisms involved. As we know, CB1 receptor signaling plays a crucial role in the ECS and pathophysiology of CHS, particularly due to its biphasic effects. At low doses of cannabinoids such as THC, CB1 receptor activation can inhibit nausea and vomiting. However, at high doses or with chronic use, CB1 receptor activation can paradoxically trigger the symptoms of CHS. This dual effect is partly attributed to receptor desensitization, internalization, and dysregulated signaling.
2. In CHS, abnormal CB1 receptor activity in both the central nervous system and peripheral tissues (e.g., gastrointestinal tract) contributes to the development of nausea and vomiting. This suggests that targeting the CB1 receptor signaling pathways, particularly through modulation of allosteric sites, could offer new therapeutic strategies for CHS intervention. Functional modulation of  GPCR signaling is very important for drug discovery, I suggest the author comment the relationship between CB1 signaling pathway and CHS.
3. Drug discovery targeting CB1 has been made much progress, like allosteric modulator as well selective orthosteric ligand. I suggest the author should comment the therapeutic application of CB1 ligands in the context of specific diseases.
4. While the manuscript describes the distinction between CHS and CVS, this section could be enhanced by further elaborating on the clinical nuances that differentiate the two conditions. For example, a deeper exploration of the role of genetic or environmental factors influencing susceptibility to CHS versus CVS would be useful for clinicians facing diagnostic challenges.

Comments on the Quality of English Language

5. There are a few minor typographical errors throughout the manuscript. For example, in the introduction section, the word “cannabis” is missing in the sentence “Cannabis has antiemetic properties at low doses…” (page 2).

Author Response

Thank you very much for taking the time to review this manuscript and providing excellent suggestions for improving it. Please find the detailed responses below and the corresponding revisions/corrections highlighted/in track changes in the re-submitted files

Response to Reviewers

First Reviewer: The manuscript provides a thorough review of Cannabis and Cannabis Hyperemesis Syndrome (CHS), from its historical background, pathophysiology, clinical presentation, and management strategies. It offers a comprehensive summary supported by various studies, which makes it a valuable resource for clinicians and researchers interested in CHS and its associated challenges. However, there are areas that require further improvement to enhance the quality of the manuscript.

Reviewer 1: the endocannabinoid system (ECS) could benefit from further clarification, especially regarding how CHS contrasts with the antiemetic effects of cannabis at lower doses. The manuscript mentions biphasic effects (antiemetic at low doses, pro-emetic at higher doses), but this concept could be expanded with more details about the molecular mechanisms involved. As we know, CB1 receptor signaling plays a crucial role in the ECS and pathophysiology of CHS, particularly due to its biphasic effects. At low doses of cannabinoids such as THC, CB1 receptor activation can inhibit nausea and vomiting. However, at high doses or with chronic use, CB1 receptor activation can paradoxically trigger the symptoms of CHS. This dual effect is partly attributed to receptor desensitization, internalization, and dysregulated signaling.

Response: Dear Reviewer, Thank you for these comments. In the pathophysiology section of the manuscript, we have additional changes to detail the endocannabinoid system (ECS) and its biphasic effects.

Reviewer 2: In CHS, abnormal CB1 receptor activity in both the central nervous system and peripheral tissues (e.g., gastrointestinal tract) contributes to the development of nausea and vomiting. This suggests that targeting the CB1 receptor signaling pathways, particularly through modulation of allosteric sites, could offer new therapeutic strategies for CHS intervention. Functional modulation of  GPCR signaling is very important for drug discovery; I suggest the author comment on the relationship between the CB1 signaling pathway and CHS.

Response: Thanks for recommending this excellent point. We have made additional changes to explain this section further in the pathophysiology and management section to incorporate the Functional modulation of  GPCR signaling and the relationship between the CB1 signaling pathway and CHS.

Reviewer 3: Drug discovery targeting CB1 has been made much progress, like allosteric modulator as well selective orthosteric ligand. I suggest the author should comment the therapeutic application of CB1 ligands in the context of specific diseases.

Response: Thanks for recommending this point. We have added comments on the therapeutic application of CB1 ligands in the context of other specific diseases. Additionally, we included a new section, “ Recent advancement,” in the management section explaining the recent progress in drug discovery targeting CB1, like allosteric modulators and selective orthosteric ligands.

Reviewer 4: While the manuscript describes the distinction between CHS and CVS, this section could be enhanced by further elaborating on the clinical nuances that differentiate the two conditions. For example, a deeper exploration of the role of genetic or environmental factors influencing susceptibility to CHS versus CVS would be useful for clinicians facing diagnostic challenges.

Response: Thanks for recommending this point. We have now included a section on genetic factors in pathophysiology. In this section, we have described the role of genetic and environmental factors in influencing the susceptibility of CHS vs. CVS.

Reviewer 5: There are a few minor typographical errors throughout the manuscript. For example, in the introduction section, the word “cannabis” is missing in the sentence “Cannabis has antiemetic properties at low doses…” (page 2).

Response: Thanks for pointing this out. We have fixed the typographical errors in the manuscript to our knowledge.

Reviewer 2 Report

Comments and Suggestions for Authors

The manuscript submitted for review provides a detailed examination of Cannabis Hyperemesis Syndrome (CHS), an increasingly recognized condition associated with chronic cannabis use. The authors have synthesized historical context, pathophysiological mechanisms, clinical presentation, and management strategies, contributing to the growing body of literature on this topic.

The abstract should be revised to provide a succinct summary of the key findings and implications. Including a statement on the clinical relevance of CHS in the context of rising cannabis use would enhance its impact.

While the historical context is informative, it could be streamlined. Focusing on key milestones directly relevant to CHS will improve clarity.

The differentiation between CHS and cyclic vomiting syndrome (CVS) is critical. A comparative table summarizing distinguishing features will aid clinicians in making accurate diagnoses.

The management section is well-detailed; however, it would benefit from referencing the latest clinical guidelines or consensus statements. Additionally, discussing the potential side effects and limitations of treatments would provide a more balanced view.

The conclusion should not only summarize the findings but also emphasize the need for ongoing research into CHS and its management. Suggestions for future research directions would add depth to this section.

Overall Recommendation: Accept with Minor Revisions

The manuscript is a valuable contribution to the understanding of CHS and will serve as a significant resource for clinicians and researchers in the field. With minor revisions to enhance clarity, update references, and improve presentation, it has the potential to make a substantial impact.

Suggested Revisions:

  1. Revise the abstract for clarity and clinical relevance.
  2. Streamline the introduction to focus on key historical points.
  3. Clarify the dual roles of the ECS in the pathophysiology section.
  4. Include a comparative table for CHS and CVS.
  5. Update the management strategies with recent guidelines and discuss potential treatment limitations.
  6. Expand the conclusion to include future research directions.

Author Response

Second Reviewer: The manuscript submitted for review provides a detailed examination of Cannabis Hyperemesis Syndrome (CHS), an increasingly recognized condition associated with chronic cannabis use. The authors have synthesized historical context, pathophysiological mechanisms, clinical presentation, and management strategies, contributing to the growing body of literature on this topic.

Reviewer 1: The abstract should be revised to provide a succinct summary of the key findings and implications. Including a statement on the clinical relevance of CHS in the context of rising cannabis use would enhance its impact.

Response: Thanks for suggesting this point. We have made changes in this section to include the clinical relevance of CHS in the context of rising cannabis use would enhance its impact.

Reviewer 2: While the historical context is informative, it could be streamlined. Focusing on key milestones directly relevant to CHS will improve clarity.

Response: Thanks for suggesting this point. After the introduction section, we have now included a figure depicting the timeline of key milestones in cannabis hyperemesis syndrome. 

Reviewer 3: The differentiation between CHS and cyclic vomiting syndrome (CVS) is critical. A comparative table summarizing distinguishing features will aid clinicians in making accurate diagnoses.

Response: Thanks for suggesting this point. Table 1 summarizes distinguishing features that will aid in making an accurate diagnosis.

Reviewer 4: The management section is well-detailed; however, it would benefit from referencing the latest clinical guidelines or consensus statements. Additionally, discussing the potential side effects and limitations of treatments would provide a more balanced view.

Response: Thanks for suggesting this timely point. We have now included a couple of statements regarding the latest clinical guidelines under the management section. We have outlined the side effects and limitations in Table 2.

Reviewer 5: The conclusion should not only summarize the findings but also emphasize the need for ongoing research into CHS and its management. Suggestions for future research directions would add depth to this section.

Response: Thanks for this suggestion. We have included additional points to emphasize future research directions in CHS and its management.

Reviewer 3 Report

Comments and Suggestions for Authors

No specific comments for Editor

Author Response

Third Reviewer: I am grateful to be invited to review the manuscript “Comprehensive review and update on Cannabis Hyperemesis Syndrome” submitted by Priyadarshini Loganathan MD and colleagues for publication in the pharmaceuticals journal. This review paper offers an in-depth overview of Cannabis Hyperemesis Syndrome (CHS), covering its pathophysiology, clinical presentation, and management strategies. It is well-organized and delves into various biological mechanisms, such as the endocannabinoid system (ECS) and the pituitary-adrenal axis. The organized layout of the manuscript ensures readers can follow the progression of CHS from its onset to treatment. One of the paper’s strong points is its detailed explanation of the pathophysiological mechanisms behind CHS. The paper deeply explains the biological processes involved by exploring the role of the endocannabinoid system (ECS) and sympathetic dysregulation. The breakdown of the ECS into ligands, receptors, and enzymes is particularly insightful. The management section is well-rounded, offering both acute and long-term treatment strategies. It covers various therapeutic approaches, including pharmacological interventions and non-pharmacological options like capsaicin and hot showers. This gives healthcare providers a range of tools to manage CHS symptoms effectively. Here are my concerns/comments that need to be addressed before this review article is published in the Pharmaceutical Journal.

Reviewer 1: Despite the importance of accurate diagnosis in differentiating CHS from similar conditions like cyclical vomiting syndrome (CVS), the review lacks a dedicated section on diagnostic tools or challenges. Including advancements in biomarkers and diagnostic criteria would add valuable practical insight.

Response: Thanks for recommending this point. We have now included a separate section for diagnosis and rearranged the information in this section. We have included statements biomarkers and diagnostic criteria for CHS.

Reviewer  2: The paper touches briefly on the rising incidence of CHS due to cannabis legalization but does not thoroughly explore the broader public health implications. Given the increasing legal and recreational use of cannabis, the review would benefit from a discussion on the role of public health campaigns and preventive strategies.

Response: Thanks for suggesting this important topic. We have now included a section on public health on page 13, emphasizing the role of public health in CHS.

Reviewer  3: While the paper mentions mood disorders in relation to long-term management, it does not explore the psychological and genetic predispositions that may contribute to CHS. Studies suggest genetic variations in cannabinoid receptors may play a role in susceptibility, yet the review underexplored this area.

Response: Thanks for pointing this out. We have now included a section on genetic variations in cannabinoid receptors and their role in CHS susceptibility.

Reviewer 4: The management section leans heavily on pharmacological therapies, with less attention given to lifestyle changes, patient education, and long-term cannabis cessation strategies. Expanding this section to cover more holistic management options would provide a more comprehensive approach to long-term care. A critical omission is the absence of any discussion of gaps in current knowledge or potential areas for future research.

Response: Thanks for pointing this out. We have now included additional psychosocial intervention in the long-term strategies.

Reviewer  5: All the botanical names/species names be written in standard italic format. The last line of the first paragraph of the introduction section, “cannabis sativa,” should be written as “Cannabis sativa.”

Response: Thanks for pointing this out. The changes are made.

Reviewer 6: The numbering of sub-sub-headings is missing. It is advisable to number all the headings and subheadings for clarity in the manuscript's structure.

Response: Thanks for informing. For clarity, we have now numbered all the headings.

Reviewer 7: Not all the references are in uniform style, such as the titles of some of the articles, which are italicized while others are not. Make sure all the references are in a uniform style.

Response: Thanks for informing. We have now arranged all references in a uniform style.